# Pyrylium based derivatization imaging mass spectrometer revealed the localization of L-DOPA

Shu Taira [1]*, Akari Ikeda[2], Yuki Sugiura[3], Hitomi Shikano[1], Shoko Kobayashi[4], Tsutomu Terauchi[2], Jun Yokoyama[2]

**1** Faculty of Food and Agricultural Sciences, Fukushima University, Fukushima, Japan, **2** Taiyo Nippon Sanso Co., Tama, Tokyo, Japan, **3** Center for Cancer Immunotherapy and Immunobiology, Kyoto University Graduate School of Medicine, Kyoto, Japan, **4** Graduate School of Agricultural and Life Sciences, The University of Tokyo, Bunkyo-ku, Tokyo, Japan

* staira@agri.fukushima-u.ac.jp

**Data Availability Statement:** All relevant data are within the paper and its Supporting Information files.

## Abstract

Simultaneous imaging of L-dihydroxyphenylalanine (L-DOPA), dopamine (DA) and norepinephrine (NE) in the catecholamine metabolic pathway is particularly useful because L-DOPA is a neurophysiologically important metabolic intermediate. In this study, we found that 2,4,6-trimethylpyrillium tetrafluoroborate (TMPy) can selectively and efficiently react with target catecholamine molecules. Specifically, simultaneous visualization of DA and NE as metabolites of L-DOPA with high steric hindrance was achieved by derivatized-imaging mass spectrometry (IMS). Interestingly, L-DOPA showed strong localization in the brainstem, in contrast to the pattern of DA and NE, which co-localized with tyrosine hydroxylase (TH). In addition, to identify whether the detected molecules were endogenous or exogenous L-DOPA, mice were injected with L-DOPA deuterated in three positions ($D_3$-L-DOPA), which was identifiable by a mass shift of 3Da. TMPy-labeled L-DOPA, DA and NE were detected at $m/z$ 302.1, 258.1 and 274.1, while their $D_3$ versions were detected at 305.0, 261.1 and 277.1 in mouse brain, respectively. L-DOPA and $D_3$-L-DOPA were localized in the BS. DA and NE, and $D_3$-DA and $D_3$-NE, all of which are metabolites of L-DOPA and $D_3$-L-DOPA, were localized in the striatum (STR) and locus coeruleus (LC). These findings suggest a mechanism in the brainstem that allows L-DOPA to accumulate without being metabolized to monoamines downstream of the metabolic pathway.

## Introduction

Neurotransmitters in the mammalian brain are known to be involved in modulating the signal transduction underlying cognitive functions, sleep and emotion. Catecholamines such as L-dihydroxyphenylalanine (L-DOPA), dopamine (DA) and norepinephrine (NE) are synthesized from tyrosine by tyrosine hydroxylase (TH), aromatic L-amino acid decarboxylase (AADC) [1] and dopamine-β-hydroxylase. Due its rapid metabolism, little is known about the localization of L-DOPA in the brain [2], which is the direct precursor of dopamine (DA) and subsequent

**Funding:** This research was funded by Japan Science and Technology (JST) Grants-in-A-STEP (VP30118067678 to S.T.), Grant-in-Aid for Scientific Research B (21H02133 to S.T.) and the LOTTE Foundation. The funders had no role in study design, data collection and analysis, decision to publish, or preparation of the manuscript.

**Competing interests:** The authors have declared that no competing interests exist.

catecholamines. Importantly, L-DOPA is used as sympomatic therapy for Parkinson's disease [3]. A clear and logical understanding of the relationships between neurotransmitters and disease requires knowledge of the spatial distribution of neurotransmitters and how it relates to the neuroanatomy of the brain. Gene expression analysis enables spatial visualization of genes involved in neurotransmitter metabolism; however, this is an indirect approach that can be difficult to correlate to neurotransmitter concentrations. Moreover, the use of immunohistochemical methods to characterize neurotransmitter localization is limited by the difficulty in generating antibodies that recognize small molecules such as catecholamines. Thus, immunological detection of small molecules is plagued by a lack of specificity for the target molecules. Although chromatography can be used to quantitatively evaluate L-DOPA concentrations, such methods are not well suited for deriving spatial information for the purpose of anatomical visualization. Functional magnetic resonance imaging (fMRI) that uses the power of magnetism to take image of body's organ and blood vessel, non-invasively [4, 5]. fMRI is powerful in clinical examination to investigate and image brain function from a signal that constructed by blood oxygenation level-dependent (BOLD) contrast [6], although an identification of molecules does not. Thus, an accurate visualization method with good spatial resolution is important for understanding monoamine metabolism. Imaging mass spectrometry (IMS) can easily recognize and provide such spatial information. Following two-dimensional MS measurements of sample sections at regular intervals, reconstruction of the target signals is obtained as an ion image. Therefore, IMS enables simultaneous detection of multiple analytes, even in the absence of target-specific markers such as antibodies [7–9], in a single experiment. To date, this method has been reported in various areas of study such as biology [10], pharmacy [9, 11], food chemistry [12–14], plant science [15, 16] and neuroscience [17–19].

In recent approaches, IMS using nucleotide as an aptamer was used for protein imaging to improve the selectivity and sensitivity [20, 21]. Because the aptamer possess high affinity towards the target molecules owing to its specific three dimensional structures [22]. In addition, tissue derivatization has been used to expand the range of molecules that can be imaged, resulting in a more versatile molecular imaging method. Although several derivatization reagents have been reported, the number of compounds to which they are applicable is limited, and more versatile derivatization reagents are required. The combination of derivatization and IMS has enabled localization of monoamines such as DA, serotonin and NE [23]. Following injection of L-DOPA to rats, DA was imaged at striatum (STR) using IMS [24], possibly indicating that exogenous L-DOPA may cross the blood brain barrier (BBB) to enter the brain and metabolized to DA.

We have also reported that a derivatization reagent with a basic pyrylium structure could be used to efficiently detect small molecules that contain primary amine groups through the reaction of pyrylium salts with nucleophiles under alkaline conditions [25, 26]. Notably, 2,4,6-trimethylpyrylium tetrafluoroborate (TMPy) selectively and efficiently reacts with the amino group of monoamines compared with other derivatization reagents that contain polyaromatic moieties (S1 File).

In this study, we used IMS with TMPy-based derivatization to clarify monoamine metabolization by identifying the brain regions where they exert their effects as well as the pathways by which they were translocated to the destination regions. Two techniques were used to investigate L-DOPA metabolism using IMS with TMPy-based derivatization. In the first, the metabolism of L-DOPA was inhibited by 3-hydroxybenzylhidrazine (NSD-1015), which is an inhibitor of AADC [27], in order to clearly image minute amounts of L-DOPA due to its rapid metabolism in the brain. The second involves deuterium-substituted L-DOPA (3-(3,4-dihydroxyphenyl-2,5,6-$D_3$)-L-alanine:$D_3$-L-DOPA) as a stable isotope compound, which was intraperitoneally (i.p.) injected to enable the discrimination of endogenous from exogenous L-

DOPA. The metabolism of other catecholamines such as DA and NE in the brain was also assessed. Application of this direct and absolute quantitative IMS method to the brain is of significant value to neurobiologists and mass spectrometrists. Therefore, visualization of L-DOPA in brain regions provides fundamental knowledge as well as advances research into cranial neuropathies.

## Material and method

### MALDI TOF-MS

The following TMPy-labeled catecholamine solutions were prepared: L-DOPA: 10 pmol/µL, DA: 10 pmol/µL and NE: 10 pmol/mL (Sigma-Aldrich, Burlington, MA, USA) and $D_3$- L-DOPA (Taiyo Nippon Sanso Co., Tokyo, Japan): 10 pmol/mL. A 2.5 µL aliquot of each sample solution was mixed with 7.5 µL of TMPy (Taiyo Nippon Sanso Co., Tokyo, Japan) (30 mM) (methanol/water/triethylamine = 70/25/5, v/v as a reaction solution) in a sealed 0.2 mL PCR test tube. The mixture was then heated at 60˚C for 10 min. A 1.0 µL aliquot of suspension containing TMPy-labeled catecholamine and CHCA (10 mg/mL) was placed on a target plate using a pipette. Ionization of TMPy-labeled L-DOPA, DA, NE and $D_3$- L-DOPA was confirmed by MALDI-TOF-MS (rapifleX, Bruker Daltonik, Bremen, GmbH). The analyte surface was irradiated with 1,000 laser shots, and TOF spectra were acquired in positive ion detection mode.

### Animal experiments

Eight-week-old female C57BL/6JJcl mice (23–25 g; Clea Japan, Tokyo, Japan) were used in accordance with the institutional Animal Experimental Guidelines of Fukushima University and approved by the Laboratory Animal Care and Use Committee of Fukushima University [Permission number, B-02]. To clearly determine the localization of L-DOPA, mice were pretreated with 3-hydroxybenzylhydrazine (Sigma-Aldrich, Burlington, MA, USA), (NSD-1015 100 mg/kg) as an inhibitor of L-aromatic amino acid decarboxylase by intraperitoneal injection (i.p.). Forty minutes following injection of NSD-1015, normal saline (control) or L-DOPA or $D_3$- L-DOPA (100 mg/kg) was individually i.p. injected. After 120 min, all mice were euthanized by cervical dislocation. The brains of these mice were dissected using surgical scissors and tweezers at room temperature, embedded into a super cryo-embedding medium (Section Lab Co., Ltd., Hiroshima, Japan), flash-frozen in liquid $N_2$ and stored at −80˚C until use. The left hemisphere was embedded in cryo-embedding medium (SECTION-LAB Co. Ltd. Hiroshima, Japan) and cut into serial sagittal sections (8 µm) using a cryostat (NX70, Thermo Fisher Scientific, Waltham, Massachusetts, USA). To investigate L-DOPA metabolism in the brain, we similarly generated a mouse model without NSD-1015.

### MALDI-TOF IMS

TMPy (4.8 mg/200 µL), (Taiyo Nippon Sanso Co., Tokyo, Japan) solution was applied to brain sections using anairbrush (Procon Boy FWA Platinum 0.2-mm caliber airbrush, Mr. Hobby, Tokyo, Japan). To enhance the reaction efficiency of TMPy on sections, the TMPy sprayed sections were placed into a dedicated container and allowed to react at 60˚C for 10 min. The container contains two channels in the central partition to wick moisture from the wet filter paper region to the sample section region. The filter paper is soaked with 1 mL of methanol/water (70/30 *v/v*) is placed next to the section inside the container. The container is then completely sealed to maintain humidity levels (S1 File). The TMPy-labeled brain sections were sprayed with matrix (CHCA-acetonitrile/water = 50/50) using an automated pneumatic sprayer

(TM-Sprayer, HTX Tech., Chapel Hill, NC, USA). Ten passes were sprayed according to the following conditions: flow rate, 120 μL/min; air flow, 10 psi; nozzle speed, 1100 mm/min.

In order to detect the laser spot area, the sections were scanned and laser spot areas (200 shots) were detected with a spot-to-spot center distance (100 μm). Signals between $m/z$ 100–1000 were corrected. The section surface was irradiated with YAG laser shots in the positive ion detection mode. The laser power was optimized to minimize in-source decay of targets. Obtained MS spectrum were reconstructed to produce MS images with a mass bin width of $m/z \pm 0.05$ from the exact mass using FlexImaging 4.0 software (Bruker Daltonics Bremen, GmbH). The peak intensity value of the spectra was normalized by dividing with the total ion current (TIC) to achieve semi-quantitative analysis between L-DOPA-treated and control mice. Optical images of brain sections were obtained by a scanner (GT-X830, Epson, Tokyo, Japan), followed by MALDI-TOF imaging MS of the sections.

## Ultra performance liquid chromatography (UPLC)

UPLC was used to quantitate the amount of L-DOPA and DA. The right brain of each mouse was divided into the brain stem, cerebellum and cerebrum (including striatum and olfactory bulb) and each region was homogenized using a ultrasonic homogenizer (Yamato Scientific co., Tokyo Japan). The obtained suspension was centrifuged to remove insoluble materials.

The supernatant of the rough extraction was purified with a spin column (MonoSpin PBA, GL Sciences, Torrance, CA, USA). The supernatant of the rough extraction was applied to the spin column and centrifuged at $10,000 \times g$ for 2 min at room temperature (RT). A 0.5 mL volume of 0.1 M HEPES buffer (pH 8.5) was applied to the spin column and centrifuged at $5,000 \times g$ for 1 min to remove any residues. Finally, 0.1 mL of 1% acetic acid was added to the spin column and centrifuged at $10,000 \times g$ for 1 min at RT to elute the target L-DOPA and dopamine.

The obtained extract was analyzed using a UPLC system (ACQUITY, Waters, USA) consisting of a binary pump, degasser, autosampler, thermostated column oven and fluorescence detector (ex/em 230/300 nm). An ODS column (ACQUITY BEH Shield RP18, 1.7 μm, $2.1 \times 50$ mm, Waters) was used. Ammonium acetate-methanol-trifluoroacetic acid (20 mM, 80:0.9:1, v/v/v) was employed as the mobile phase at a flow rate of 0.5 mL/min and run time of 5 min.

## Results and discussion

### TMPy-labeled monoamines

All analyses were performed six times to confirm the reproducibility of the technique. The detected masses of TMPy-labeled standard L-DOPA ($m/z$ 302.1), $D_3$-L-DOPA ($m/z$ 305.1), DA ($m/z$ 258.1) and NE ($m/z$ 274.1) increased by 105.0 Da compared with the original masses (Mw 197.1, 200.1, 153.1 and 169.1). Tandem MS confirmed the fragmentation ions of TMPy from the standard sample (Table 1). A fragmented ion of the pyridine ring moiety ($m/z$ 122.1) was regularly cleaved and observed for all TMPy-modified target molecules.

As well as standard target result, TMPy could react to catechol amins on section (Table 2 and S1 File). From section results, DA and $D_3$-DA, and NE and $D_3$-NE could confirm indicating that injected-L-DOPA and -$D_3$-L-DOPA metabolized into DA and $D_3$-DA, and NE and $D_3$-NE in brain. All detected precursor ions showed TMPy-fragmented ion ($m/z$ 122.1), although NE and $D_3$-NE did not detect main body ions ($m/z$ 153.1 and 156.1) due to may lower metabolism from DA into NE. The detection of both precursor and fragmented TMPy unequivocally demonstrates the modification of the target molecule by TMPy.

**Table 1. Selected monitor ions and detection limit for TMPy-labelled standard catechol amines.**

| Target | Precursor ($m/z$) | Fragment ($m/z$) | LOD (pmol) |
|---|---|---|---|
| L-DOPA | 198.1 | 181.1 | 24.7 |
| $D_3$-L-DOPA | 201.1 | 184.0 | 42.4 |
| Dopamine (DA) | 154.1 | 137.0 | 221 |
| Noradrenaline (NE) | 170.2 | 153.1 | 87.7 |
| TMPy-L-DOPA | 302.1 | 122.1, 181.1 | 0.72 |
| TMPy-$D_3$-L-DOPA | 305.1 | 122.1, 184.0 | 0.50 |
| TMPy-DA | 258.1 | 122.1, 137.0 | 5.90 |
| TMPy-NE | 274.1 | 122.1, 153.1 | 0.05 |

All TMPy derivatized target molecules can be aligned in positive mode, which suggested measurements in both polarities are not required even so original target ion polarities is negative. These results demonstrated the utility of TMPy in improving ionization efficiency. A summary of the target molecules for detected ions and limits of detection (LOD) is shown in Table 1. LOD was calculated from Eq (1).

$$LOD = 3.3s/a, \qquad (1)$$

where s is the standard division of the signal-to-noise ratio and a is the slope of the calibration curve.

The structure of most conventional derivatization reagents consists of a polyaromatic molecule [24]. Thus, the primary amino group that binds to short alkyl chains like L-DOPA cannot efficiently react with conventional derivatization reagents. Importantly, TMPy was able to react with the amine group of L-DOPA without steric hindrance, suggesting that TMPy is one of a few reagents that can efficiently detect small molecule.

TMPy has another advantage in tandem MS imaging. We were able to regularly detect fragmented TMPy ($m/z$ 122.1) from TMPy-labeled targets, which provides supporting evidence for the detection of derivatized target molecules in MS mode. Candidate $m/z$ for target catecholamines indicated TMPy ($m/z$ 122.1) in the brain, suggesting successful Py-labeling on the tissue section. For example, when we imaged dopamine at $m/z$ 258.1 (TMPy-DA candidate), the tandem MS image for $m/z$ 122.1 (precursor ion at $m/z$ 258.1) clearly showed a corresponding localization (S1 File).

## UPLC for L-DOPA and DA in NSD-1015-treated mouse brain

The brain was divided into the cortex, cerebellum and brain stem (BS), and L-DOPA and DA (see experimental section) were extracted from each brain region. L-DOPA and DA standards were separated under the same gradient conditions. In this experiment, $D_3$-L-DOPA was converted using the L-DOPA standard. We selected 2 different peaks that correspond to L-DOPA

**Table 2. Selected monitor ions and detection limit for TMPy-labelled catechol amines on brain.**

| Target | Precursor ($m/z$) | Fragment ($m/z$) |
|---|---|---|
| TMPy-L-DOPA | 302.1 | 122.1, 181.1 |
| TMPy-$D_3$-L-DOPA | 305.1 | 122.1, 184.0 |
| TMPy-DA | 258.1 | 122.1, 137.0 |
| TMPy-$D_3$-DA | 261.1 | 122.1, 140.1 |
| TMPy-NE | 274.1 | 122.1 |
| TMPy-$D_3$-NE | 277.1 | 122.1 |

(retention time, RT: 0.5 min) and DA (RT: 0.9 min). These retention times are reasonable for reversed phase LC because of the dependence on molecular hydrophobicity, which is affected by the presence/absence of carboxyl groups on the side chain. The targets were quantified using a standard method based on peak area. The amounts of L-DOPA were $1.1 \pm 0.2$, $0.8 \pm 0.2$, $3.6 \pm 0.6$ and $4.0 \pm 0.1$ ng/mg in the cortex; and $1.9 \pm 0.1$, $2.5 \pm 0.96$, $4.3 \pm 4.2$ and $4.7 \pm 0.5$ ng/mg in the cerebellum and $3.9 \pm 1.0$, $9.3 \pm 2.3$, $100 \pm 1.9$ and $77 \pm 6.5$ ng/mg in the BS for the control, NSD-1015-treated, (NSD-1015 and L-DOPA)-treated and (NSD-1015 and $D_3$-L-DOPA)-treated mice, respectively. In a comparison of the cortex and cerebellum regions, the amount of L-DOPA showed no significant difference regardless of the experimental condition. In the BS, significantly elevated levels of L-DOPA were observed in the (NSD-1015 and L-DOPA)-treated and (NSD-1015 and $D_3$-L-DOPA)-treated mice compared to the control and NSD-1015-treated mice ($P<0.01$) (Fig 1A). On the other hand, the highest levels of DA ($86 \pm 0.2$ ng/mg) were observed in the cortex of control mice compared to the other regions and experimental conditions (i.e., 1–4.5 ng/mg) (Fig 1B), indicating that normal metabolism from L-DOPA to DA progressed in the brain. This result indicates that NSD-1015 treatment directly affected the amount of L-DOPA in the brain. The amount of L-DOPA was clearly elevated in the BS, suggesting that the L-DOPA firstly accumulate at BS due to inhibition of metabolism by NSD-1015. To prove this result, we visualized the localization of monoamines by TMPy derivatization-based imaging MS.

## TMPy derivatization-based imaging MS

**NSD-1015 inhibition of L-DOPA metabolism.** In this experiment, the exact location of monoamines was investigated with the use of TMPy derivatization-based imaging MS. TMPy derivatization of all target signals was confirmed by tandem MS at *m/z* 302.1, 305.1 and 258.1 in tissue sections and showed a pyrylium signal (*m/z* 122.1) correlated with the fragmented ion of TMPy-derivatized L-DOPA, $D_3$-L-DOPA and DA, respectively. The obtained imaging data were normalized to total ion counts. From all samples (control (Fig 2A), NSD-1015-treated

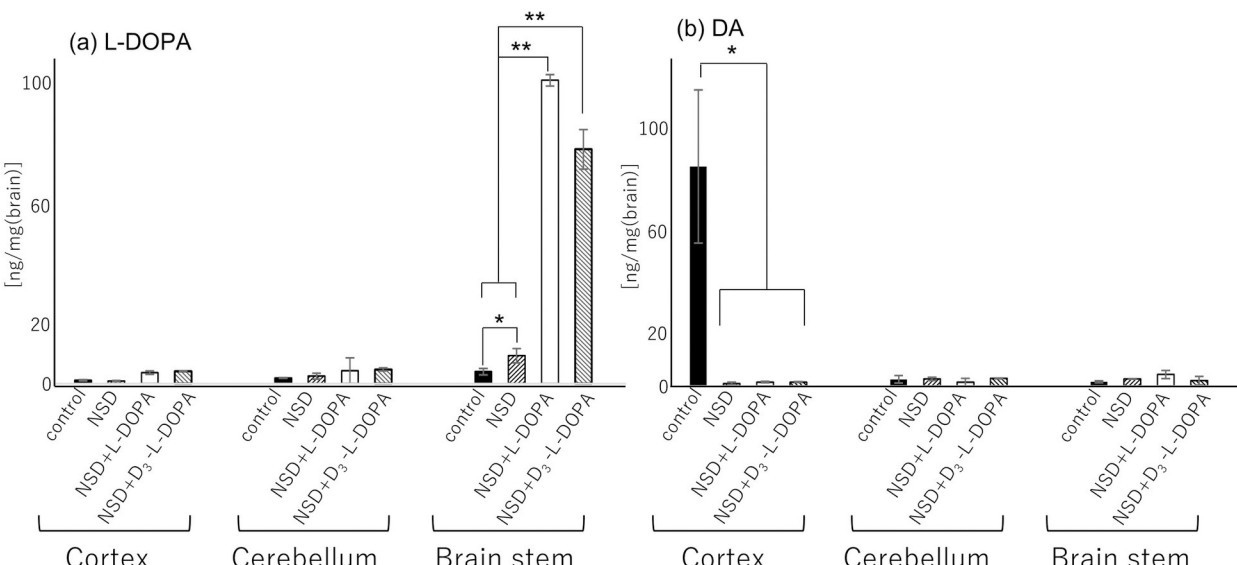

**Fig 1.** Quantitative comparison of the amount of L-DOPA (a) and DA (b) in control, NSD-, NSD and L-DOPA-, and $D_3$-L-DOPA and NSD-treated mice (B). Obtained values were calculated per unit area. The values are expressed as mean + SEM. *p < 0.05 and **p < 0.01 with student's t-test. n = 3.

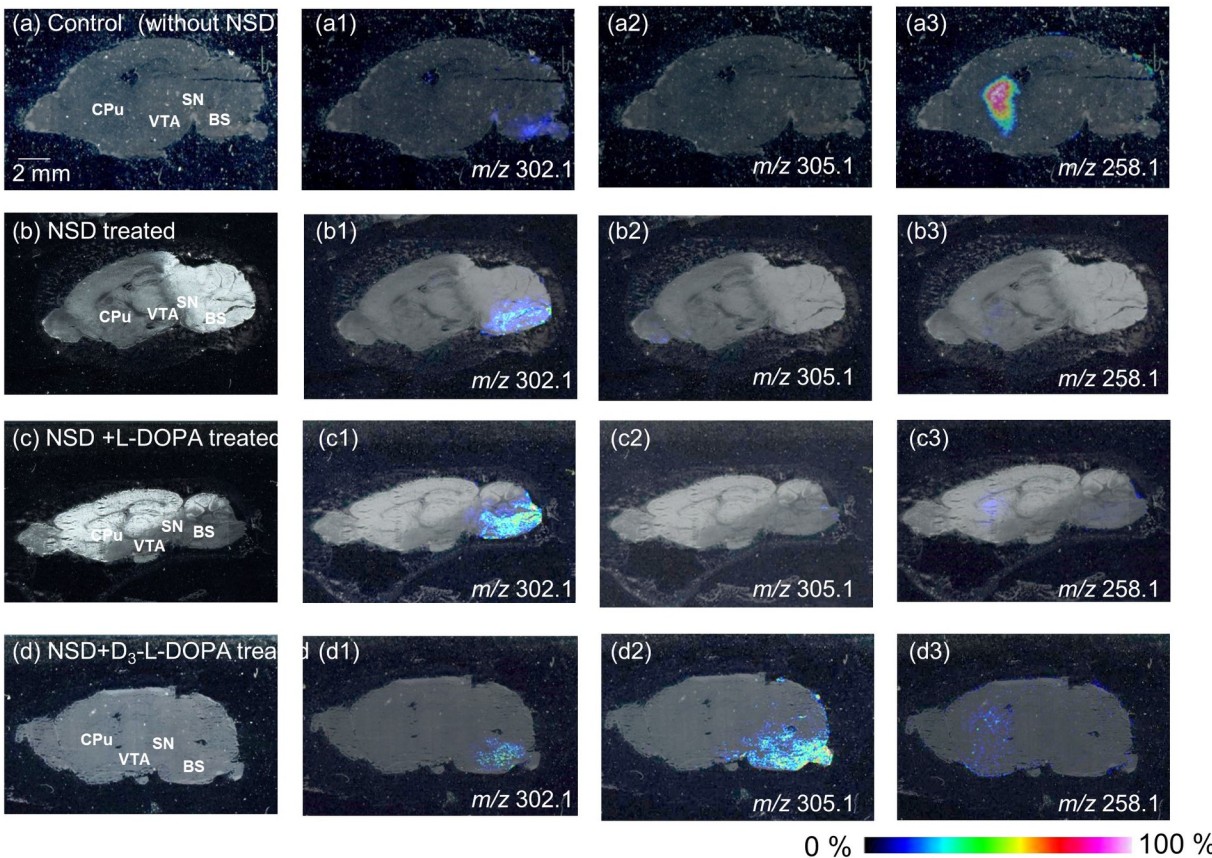

**Fig 2. Derivatized imaging mass spectrometry of catechol amines.** Optical image of sagittal section from mouse brain saline-treated as control (a), NSD-1015-treated (b), NSD1015 and L-DOPA-treated (c) and NSD1015 and $D_3$-L-DOPA-treated (d) mouse. MS spectra reconstructed image of L-DOPA (a1, b1, c1 and d1), $D_3$-L-DOPA (a2, b2, c2 and d2) and DA (a3, b3, c3 and d3), BS, brain stem. MY, medulla; OB, olfactory bulb; P, Pons.

(Fig 2B), NSD-1015 and L-DOPA-treated (Fig 2C) and NSD-1015 and $D_3$-L-DOPA-treated mice (Fig 2D)), TMPy-labeled L-DOPA (*m/z* 302.1) was detected and mainly imaged in the brain stem (BS), a region that is related to sensory function and behavior (Fig 2A1, 2B1, 2C1 and 2D1), although differences in the image contrast were observed. Relative comparisons of total ion counts for L-DOPA at the BS indicated ratios of 0.2:1:2.1:0.8 (control: NSD-1015-: NSD-1015 and L-DOPA-: NSD-1015 and $D_3$-L-DOPA-treated mice). The image was clearer when NSD-1015 was treated (Fig 2B1) compared to the control (Fig 2A1), indicating that NSD-1015 could inhibit L-DOPA metabolism. In addition, L-DOPA levels were increased following L-DOPA treatment and showed the highest intensity in the BS (Fig 2C1) compared to the other regions. The image of TMPy-$D_3$-L-DOPA (*m/z* 305.1) was confirmed in only the $D_3$-L-DOPA-treated brain and showed localization in the BS region (Fig 2D2), indicating that the injection of exogenous $D_3$-L-DOPA could cross the BBB and enter the brain. For L-DOPA from both L-DOPA- and $D_3$-L-DOPA-injected brains, we found that L-DOPA was delivered to and preferentially accumulated in the BS which is in good agreement with the UPLC data (Fig 1).

DA as a metabolite of L-DOPA was also imaged in the Caudate-Putamen (CPu) under all experimental conditions (Fig 2A3, 2B3, 2C3 and 2D3). The control brain showed the highest contrast (Fig 2A3) compared with the others. In groups treated with NSD-1015, DA was

hardly detected in the brain due to NSD-1015 inhibition of L-DOPA metabolism to DA (Fig 2B3).

It is known that L-DOPA was rapidly decarboxylated and formed DA in the substantia nigra (SN) by the action of aromatic L-amino acid decarboxylase (AADC), and was transported to the CPu. Until now, little is known about the localization of L-DOPA in the brain [2]. Immunostaining method is common approach to determine localization biotargets. Okamura et al. showed that L-DOPA-immunoreactive (IR) neurons are located in the ventral tegmental area (VTA) but not in the STR. In contrast, DA could be detected in the VTA and STR. They hypothesized that L-DOPA was below the detection limit of the immunological technique in the other regions [1]. As other target to evaluate catecholamines, Tyrosine hydroxylase (TH) is the rate-limiting enzyme in DA synthesis and is known to localize in the STR and to be absent in the BS [28–30]. Thus, it is difficult to visualize L-DOPA in the brain. Our data suggested that while IR and TH can be used to evaluate DA synthesis, TMPy derivatization-based IMS can be used to evaluate L-DOPA localization by direct molecular imaging. To determine optimized reaction condition of TMPy, different TMPy reaction conditions with the targets were considered. One is imaging experiment has achieved without incubation. Target molecules did not detect and image clearly without incubation, indicating that TMPy needed to be under moist state in order to react with target molecules. The other one is imaging experiment has achieved with methanol/ water (30/70 *v/v*) condition. Under this condition, obscure image was obtained due to that target molecule has migrated from original position due to moist-state (data not shown). Thus, our suggested condition (methanol/water (70/30 *v/v*) solution at 60˚C. for 10 min.) is most suitable to react TMPy derivatization reagent on section.

**Non-inhibited L-DOPA metabolism.** To visually evaluate the metabolism of L-DOPA or $D_3$-L-DOPA in the brain, L-DOPA- (Fig 3A–3G) or $D_3$-L-DOPA- (Fig 3H–3N) treated mouse brains without NSD-1015 were assessed. For normal mice without NSD-1015, L-DOPA was hardly imaged in the brain sample (Fig 2A1). Thus, in this experiment, L-DOPA-treated mice were the control. TMPy-L-DOPA (*m/z* 302.1) was localized in the BS in both L-DOPA-injected and D3-L-DOPA-injected brains (Fig 3B and 3I). This was similar to the distribution observed following NSD-1015 treatment (Fig 2), although a slight decrease was observed. The relative intensities observed in (NSD-1015 and L-DOPA)-injected (Fig 2B1), L-DOPA-injected (Fig 3B) and $D_3$-L-DOPA-injected brains (Fig 3I) were 4:1:1.1. Notably, the localization of L-DOPA between L-DOPA-injected and $D_3$-L-DOPA-injected brains showed some differences. The BS is divided into two regions, the pons (P) and medulla (MY)). L-DOPA localized in the entire BS region in the L-DOPA-injected mice (Fig 3B). On the other hand, $D_3$-L-DOPA-injected mice exhibited L-DOPA localization in only the P in the BS (Fig 3I). No localization of L-DOPA in the MY was observed. The observed difference between L-DOPA- and $D_3$-L-DOPA-injected mice is attributable to the presence of exogenous L-DOPA. We hypothesized that exogenous L-DOPA was first delivered to the MY through the medulla spinalis, and subsequently translocated from the MY to the P. Thus, $D_3$-L-DOPA-injected mice showed no localization of L-DOPA in the MY as a result of the lack of an endogenous supply of L-DOPA.

In addition, we were able to image TMPy-$D_3$-L-DOPA (*m/z* 305.1) at the boundaries between the P and MY, rather than being restricted to the P region (Fig 3J), which represents an image of $D_3$-L-DOPA being transported from the MY to P and supports our hypothesis.

The striatum (STR) is divided into two regions, the caudate putaman (CPu) and nucleus accumbens (NAcc). TMPy-DA (*m/z* 258.1) was able to be imaged and was localized in the STR (both CPu and NAcc) in L-DOPA-treated mice (Fig 3D). In $D_3$-L-DOPA-treated mice, DA was weakly imaged in the STR (Fig 3K). Relative comparison of total ion counts for DA in the STR indicated 1.8:1 (L-DOPA-treated mice: $D_3$-L-DOPA-treated mice) due to treatment with exogenous L-DOPA. In contrast, TMPy-$D_3$-DA (*m/z* 261.1) was mainly imaged in the CPu in the

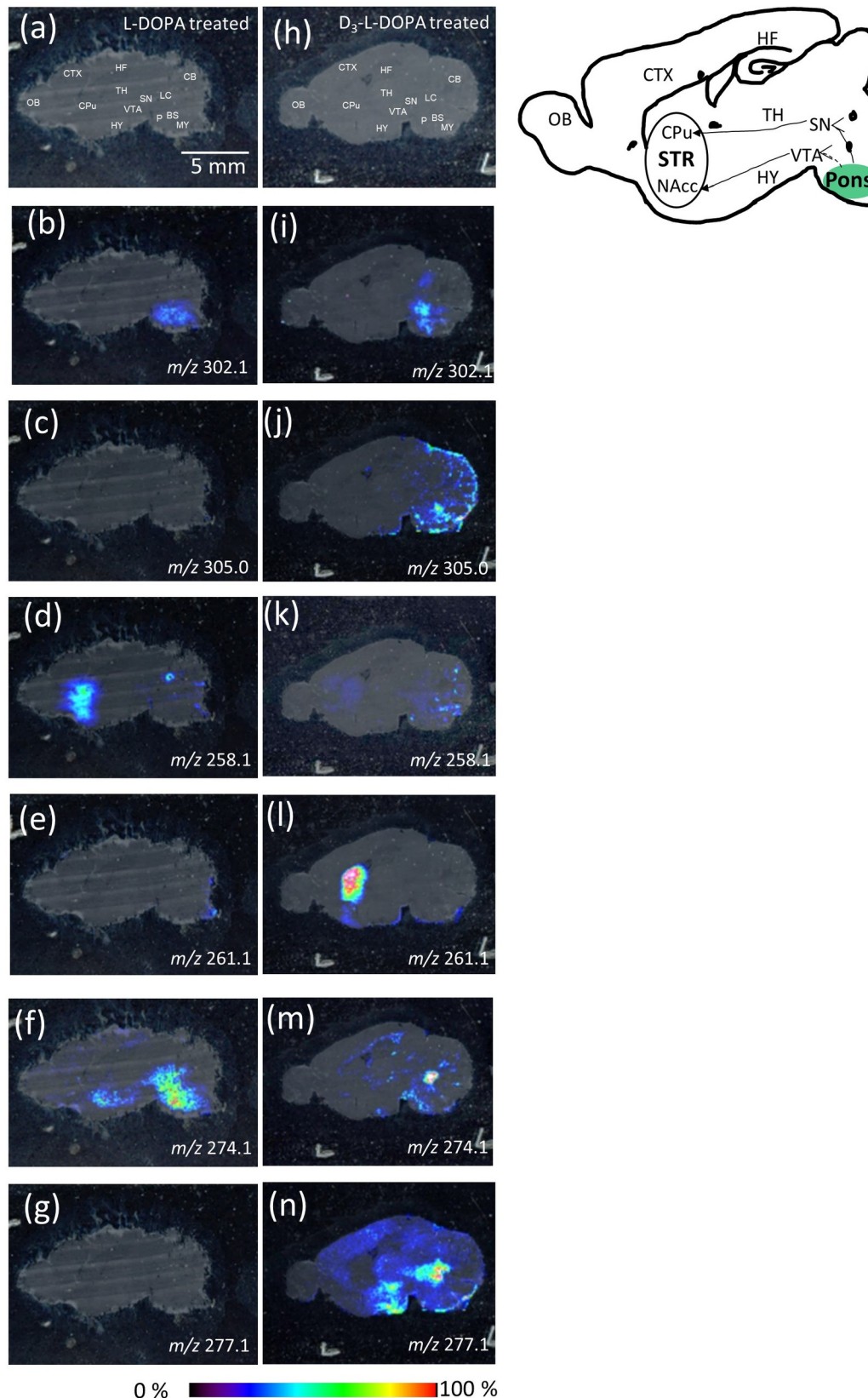

**Fig 3. Derivatized imaging mass spectrometry of catechol amines.** Optical image of sagittal section from mouse brain (a) L-DOPA-treated and (h) $D_3$-L-DOPA-treated mouse. MS spectra reconstructed image of L-DOPA (b) L-DOPA-treated and (i) $D_3$-L-DOPA-treated mouse, D3-L-DOPA (c) L-DOPA-treated and (j) $D_3$-L-DOPA-treated mouse, dopamine (DA) (d) L-DOPA-treated and (k) $D_3$-L-DOPA-treated mouse, $D_3$-DA (e) L-DOPA-treated and (l) $D_3$-L-DOPA-treated mouse, norepinephrine (NE) (f) L-DOPA-treated and (m) $D_3$-L-DOPA-treated mouse and $D_3$-NE (g) L-DOPA-treated and (n) $D_3$-L-DOPA-treated mouse. BS, brain stem CB, cerebellum, Striatum, STR, CTX, cerebral cortex, HY, hypothalamus, HF, hippocampal formation, LC, locus coeruleus, MY, medulla, OB, olfactory bulb, P, Pons, SN, substantia nigra, TH, thalamus, VTA; ventral tegmental area.

$D_3$-L-DOPA-treated mice (Fig 3L), indicating that $D_3$-L-DOPA is metabolized to $D_3$-DA in the brain. The relative intensity of $D_3$-L-DOPA increased by about 1.4-fold in the $D_3$-L-DOPA-treated mice compared with DA in the L-DOPA-treated mice (Fig 3D). We discussed for a projection pathway of DA from IMS data. Generally, the CPu and NAcc receives DA input primarily from the SN and ventral tegmental area (VTA), respectively. The DA which received from SN and VTA differentially works to action or reward forecast, respectively. From IMS data, DA localized at whole STR indicating that treated-L-DOPA transported from both VTA and SN (Fig 3D). In contrast, $D_3$-DA was mainly imaged at CPu, indicating $D_3$-L-DOPA preferentially transported to SN. Subsequently, $D_3$-DA which was D3-L-DOPA metabolite transported to CPu from SN (Fig 3L). Thus, we hypothesized $D_3$-L-DOPA may correlates with an action although we should examine this phenomenon mechanism using other animal model (ie. Parkinson's disease model). To assess subsequent metabolism of DA to NE, TMPy-derived NE was detected at *m/z* 274.1 and was localized in the local coeruleus (LC), hypothalamus (HY) and P in the brains of L-DOPA-treated mice (Fig 3F). $D_3$-NE was not imaged in the L-DOPA-treated mouse brain (Fig 3G). It is known that NE projections to the HY are involved in cognitive function. NE projects to the P and MY, regions that are adjacent to adrenergic neurons [31], which suggests that NE was preferentially used as a precursor of adrenaline when there was an abundance of NE in the brain. Adrenaline in the MY subsequently forms descending projections that regulate the sympathetic nerve. In fact, $D_3$-L-DOPA-treated mice showed localization of NE only in the LC due to the lack of excess L-DOPA, as observed for L-DOPA treated mice (Fig 3M). Additionally, $D_3$-NE (*m/z* 277.1), which is a metabolite of $D_3$-DA, was imaged in the LC, thalamus (TH), HY and cerebral cortex (CTX). Interestingly, the relative intensity of $D_3$-NE increased by about 1.7-fold in $D_3$-L-DOPA-treated mice compared with NE in L-DOPA-treated mice due to the presence of several projection regions (Fig 3F and 3N). The projection to several regions could indicate that deuterated NE was used a neurotransmitter in cognitive function [32], escape of stress [33] and awareness [34] rather than as a precursor of adrenaline.

In conclusion, our findings demonstrated the localization of L-DOPA and its metabolites using TMPy derivatization-based imaging mass spectrometry technique. Due to improvements in LOD using the TMPy reagent, we were able to observe the localization and metabolism of catecholamines in the absence of NSD-1015. Deuterium-substituted L-DOPA was also imaged and enabled the differentiation between endogenous and exogenous L-DOPA in the brain. Moreover, deuterated L-DOPA was metabolized *in vivo* and formed deuterated DA and NE in the brain. Interestingly, the terminal region of neuronal projections differed for the metabolites of $D_3$-L-DOPA such as $D_3$-DA and $D_3$-NE compared with that of L-DOPA. In addition, image intensity of deuterated DA (Fig 3L) and NE (Fig 3N) were higher than normal DA (Fig 3D) and NE (Fig 3F) suggesting that turnover of deuterated L-DOPA was faster that of normal L-DOPA, and metabolites were preferentially located in specific regions that are correlated with movement and cognition function. Further investigation is needed to clarify the function of deuterated L-DOPA and metabolites in the brain. Thus, in the future we will investigate the effect of deuterated L-DOPA on neurotransmitter function using animal models of cranial neuropathies.

## Supporting information

**S1 File.**
(ZIP)

## Author Contributions

**Conceptualization:** Shu Taira.

**Data curation:** Shu Taira.

**Formal analysis:** Shu Taira, Yuki Sugiura.

**Funding acquisition:** Shu Taira.

**Investigation:** Shu Taira, Akari Ikeda, Yuki Sugiura, Shoko Kobayashi, Tsutomu Terauchi, Jun Yokoyama.

**Methodology:** Shu Taira, Akari Ikeda, Yuki Sugiura, Hitomi Shikano.

**Project administration:** Shu Taira, Jun Yokoyama.

**Supervision:** Jun Yokoyama.

**Visualization:** Shu Taira.

**Writing – original draft:** Shu Taira.

**Writing – review & editing:** Shu Taira.

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
