## [Decision Letter · Decision Letter 0]

10 May 2022

PONE-D-22-05574Pyrylium based derivatization imaging mass spectrometer revealed the localization of L-DOPAPLOS ONE

Dear Dr. Taira,

Thank you for submitting your manuscript to PLOS ONE. After careful consideration, we feel that it has merit but does not fully meet PLOS ONE’s publication criteria as it currently stands. Therefore, we invite you to submit a revised version of the manuscript that addresses the points raised during the review process.

We look forward to receiving your revised manuscript.

Kind regards,

Jonghoon Choi, Ph.D.

Academic Editor

PLOS ONE

Journal Requirements:

“This research was funded by Japan Science and Technology (JST) Grants-in-A-STEP (VP30118067678 to S.T.), Grant-in-Aid for Scientific Research B (21H02133 to S.T.) and the LOTTE Foundation.”

Reviewers' comments:

Reviewer's Responses to Questions

**Comments to the Author**

1. Is the manuscript technically sound, and do the data support the conclusions?

Reviewer #1: Yes

Reviewer #2: Yes

2. Has the statistical analysis been performed appropriately and rigorously? 

Reviewer #1: Yes

Reviewer #2: Yes

3. Have the authors made all data underlying the findings in their manuscript fully available?

Reviewer #1: Yes

Reviewer #2: No

4. Is the manuscript presented in an intelligible fashion and written in standard English?

Reviewer #1: Yes

Reviewer #2: Yes

5. Review Comments to the Author

Reviewer #1: Was the TMPy-derivatation reasonably enough? Please show if there are any evidence of heating the mixture for 60℃, 10min.

Between each sample, how much if there were any difference between the efficiency of TMPy-derivatation?

The elimination half-time of L-DOPA is around 0.75 to 1.5 hours. How will the pharmacodynamics change with the administration of NSD-1015, before euthanasia?

Reviewer #2: - Authors introduced a unique way to image the brain's neurotransmitters by mass spectroscopy. The localization of the concentrations of the neurotransmitters were achieved by nucleophilic reaction of pyrylium structures. The topics were scientifically designed and results have shown meaningful brain images. However, there are issues which should be resolved before consideration of publication.

1. IMS sounds very powerful to surpass the capabilities based on immuno or immunochemical binding methods, which lacks specificity. Then, how about other bioprobes such as recent one, aptamer?

2. TMPy labeling on brain section should accompany unwanted monoamine compounds in brain. Authors should describe how the background reaction from the TMPy could be removed or minimized.

3. There are no synergistic aspects other than endogeneous or exogeneous neurotransmitters between two methods of TMPy labeling and deutrium-substituted neurotransmitters?

4. The process of TMPy labeling need a relatively high temperature of 60oC. Is there any other available temperature around mouse's vital tone?

5. Introduction needs broader range of descriptions about brain imaging or brain mapping, for example, including fMRI, ETC.

6. PLOS authors have the option to publish the peer review history of their article (what does this mean?). If published, this will include your full peer review and any attached files.

Reviewer #1: No

Reviewer #2: No

---

## [Author Response · Author response to Decision Letter 0]

13 Jun 2022

Thank you for giving us an opportunity to re-submit our manuscript, formerly entitled “Pyrylium based derivatization imaging mass spectrometer revealed the localization of L-DOPA”, to PLOS ONE.

We thank you and the reviewers for helpful suggestions for improving our paper. In response to the reviewers’ comments, we feel that the manuscript is much improved because of them.

We hope the responses and the revised manuscript will meet your criteria for publication in PLOS ONE.

I would like to thank you in advance for considering this manuscript. Please feel free to contact me if you have any questions or require further information. 

Sincerely,

Shu Taira, PhD

Professor 

Response sheet Reviewer #1

1) Reviewer #1: Was the TMPy-derivatation reasonably enough? Please show if there are any evidence of heating the mixture for 60℃, 10min. 

Between each sample, how much if there were any difference between the efficiency of TMPy-derivatation?

A. Thank you for your comments. To determine optimized reaction condition of TMPy, different TMPy reaction conditions with the targets were considered. One is imaging experiment has achieved without incubation. Without incubation, target molecules did not detect and image clearly, indicating that TMPy needed to be under moist state in order to react with target molecules. 

The other one is imaging experiment has achieved with methanol/ water (30/70 v/v) condition. Under this condition, target molecule has migrated from original position due to moist-state.Thus, our suggested condition (methanol/water (70/30 v/v) solution at 60 oC. for 10 min.) is most suitable to react TMPy derivatization reagent on section. To avoid misunderstanding for broad reader, we added this result in revised manuscript (red character).

2) The elimination half-time of L-DOPA is around 0.75 to 1.5 hours. How will the pharmacodynamics change with the administration of NSD-1015, before euthanasia?

A. Thank you for your advice. In this time, L-DOPA was not affected by an elimination half-time. Because forty minutes following injection of NSD-1015, L-DOPA or D3- L-DOPA (100 mg/kg) was individually i.p. injected. After 120 min, all mice were euthanized by cervical dislocation. Thus, an exogenous L-DOPA or D3-L-DOPA marginally metabolized (eliminated half-time). Therefore, exogenous L-DOPA delivered and accumulated without metabolism at brain stem (BS). In the previous manuscript (Line 127-129 ), our sentences was misleading for experimental procedure. Thus, we revised in this section. 

Response sheet for Reviewer #2

Reviewer #2: - Authors introduced a unique way to image the brain's neurotransmitters by mass spectroscopy. The localization of the concentrations of the neurotransmitters were achieved by nucleophilic reaction of pyrylium structures. The topics were scientifically designed and results have shown meaningful brain images. However, there are issues which should be resolved before consideration of publication.

1. IMS sounds very powerful to surpass the capabilities based on immuno or immunochemical binding methods, which lacks specificity. Then, how about other bioprobes such as recent one, aptamer?

A. Thank you for kind suggestion. There are some reports for novel imaging method using aptamer. Most of the reports involve fluorescent reagent or nanoparticle aptamers modified with nucleic acids. Certainly, these results were selective and sensitive. Somehow, we described IMS advantages compared with aptamer imaging, IMS could image small molecules such as catechol amines, and gives us exhaustive data for omics analysis. Thus, we may find biomarker molecules from IMS data. In revised manuscript, we added new references for aptamer imaging as other bioimaging method. 

2. TMPy labeling on brain section should accompany unwanted monoamine compounds in brain. Authors should describe how the background reaction from the TMPy could be removed or minimized.

A. Thank you for your comment. IMS selects one m/z ± 0.05 signal and image target molecules. Thus, other unwanted monoamine which is difference m/z compared with that of target did not image as background. In this paper we imaged known monoamines which knew localization area. (ie. Dopamine localized caudate putamen (CPu) in STR.) 

 To avoid ion suppression by unwanted monoamines, we confirmed ion intensity at several section points as pre-measurement. In pre-measurement, we set laser power (<60 J/m2) to avoid over saturated ion count. 

3. There are no synergistic aspects other than endogeneous or exogeneous neurotransmitters between two methods of TMPy labeling and deutrium-substituted neurotransmitters?

A. Thank you for your question. TMPy labeling enhanced detection limit (Support information 1). In present manuscript, we described the transportation route of L-DOPA (Line 308-322) and dopamine by endogeneous or exogeneous L-DOPA . In addition, we re-discussed the effect deuterium L-DOPA. Striatum (STR) was divided into two regions, caudate putaman (CPu) and nucleus accumbens (NAcc). Generally, the CPu and NAcc receives DA input primarily from the SN and ventral tegmental area (VTA), respectively. The DA which received from SN and VTA differentially works to action or reward forecast, respectively. From IMS data, DA localized at whole STR indicating that treated-L-DOPA transported from both VTA and SN (Fig. 3d). In contrast, D3-DA was mainly imaged at CPu, indicating D3-L-DOPA preferentially transported to SN. Subsequently, D3-DA which was D3-L-DOPA metabolite transported to CPu from SN (Fig. 3l). Thus, we hypothesized D3-L-DOPA may correlates with an action although we should examine this phenomenon mechanism using other animal model (ie. Parkinson’s disease model). We added discussion in revised manuscript (Line 338-347) and illustration of projection of catechol amine in brain in Figure 3. 

4. The process of TMPy labeling need a relatively high temperature of 60oC. Is there any other available temperature around mouse's vital tone?

A. Thank you for kind suggestion. We have tried to react TMPy in other condition. The imaging experiment has achieved without incubation. Target molecules did not detect and image clearly, indicating that TMPy could not react with target molecules under room temperature. The merit of reaction at 60 oC is a vapor condition. To enhance the reaction efficiency of TMPy on sections, the TMPy sprayed sections were placed into a dedicated container and allowed to react at 60 oC for 10 min. The container contains two channels in the central partition to wick moisture from the wet filter paper region to the sample section region. The filter paper is soaked with 1 mL of methanol/water (70/30 v/v) or methanol/water (30/70 v/v) and is placed next to the section inside the container. The container is then completely sealed to maintain humidity levels. Resulting, target monoamines detected and imaged in methanol/water (70/30 v/v) condition. In the case of methanol/water (30/70 v/v) that is high humidity, target molecules could react to TMPy, although obscure image was obtained due to that a migration of target molecules on section occurred. Thus, we adopted the condition in methanol/water (70/30 v/v) at 60 oC for 10 min. we revised manuscript in material and method (Line 140-146) and Line294-303, and added the information of dedicated container in supporting information 2.

5. Introduction needs broader range of descriptions about brain imaging or brain mapping, for example, including fMRI, ETC.

A. Thank you for your kind suggestion. As other imaging method, we cited fMRI in revised manuscript. Due to this citation, we clearly described the differentiation of character of IMS.

---

## [Decision Letter · Decision Letter 1]

6 Jul 2022

Pyrylium based derivatization imaging mass spectrometer revealed the localization of L-DOPA

PONE-D-22-05574R1

Dear Dr. Taira,

We’re pleased to inform you that your manuscript has been judged scientifically suitable for publication and will be formally accepted for publication once it meets all outstanding technical requirements.

Kind regards,

Jonghoon Choi, Ph.D.

Academic Editor

PLOS ONE

Additional Editor Comments (optional):

Reviewers' comments:

Reviewer's Responses to Questions

**Comments to the Author**

1. If the authors have adequately addressed your comments raised in a previous round of review and you feel that this manuscript is now acceptable for publication, you may indicate that here to bypass the “Comments to the Author” section, enter your conflict of interest statement in the “Confidential to Editor” section, and submit your "Accept" recommendation.

Reviewer #1: All comments have been addressed

Reviewer #2: All comments have been addressed

2. Is the manuscript technically sound, and do the data support the conclusions?

Reviewer #1: Yes

Reviewer #2: Yes

3. Has the statistical analysis been performed appropriately and rigorously? 

Reviewer #1: Yes

Reviewer #2: I Don't Know

4. Have the authors made all data underlying the findings in their manuscript fully available?

Reviewer #1: Yes

Reviewer #2: Yes

5. Is the manuscript presented in an intelligible fashion and written in standard English?

Reviewer #1: Yes

Reviewer #2: Yes

6. Review Comments to the Author

Reviewer #1: (No Response)

Reviewer #2: 'Cause entire issues having numer from 1 to 4 raised from this reviewer were resolved, it can be acceptable.

7. PLOS authors have the option to publish the peer review history of their article (what does this mean?). If published, this will include your full peer review and any attached files.

Reviewer #1: No

Reviewer #2: No

---

## [Editor Report · Acceptance letter]

25 Jul 2022

PONE-D-22-05574R1 

Pyrylium based derivatization imaging mass spectrometer revealed the localization of L-DOPA 

Dear Dr. Taira:

I'm pleased to inform you that your manuscript has been deemed suitable for publication in PLOS ONE. Congratulations! Your manuscript is now with our production department. 

Kind regards, 

on behalf of

Prof. Jonghoon Choi 

Academic Editor

PLOS ONE